# Evaluation of a Programme of Online Arts Activities for Patients with Kidney Disease during the COVID-19 Pandemic

**DOI:** 10.3390/healthcare10020260

**Published:** 2022-01-28

**Authors:** Anna Wilson, Claire Carswell, Stephanie Burton, William Johnston, Jennifer Baxley Lee, Alistair MacKenzie, Michael Matthews, Paul Murphy, Joanne Reid, Ian Walsh, Fina Wurm, Helen Noble

**Affiliations:** 1School of Nursing and Midwifery, Queen’s University Belfast, Belfast BT7 1NN, UK; c.carswell@qub.ac.uk (C.C.); sburton06@qub.ac.uk (S.B.); mmatthews01@qub.ac.uk (M.M.); p.murphy@qub.ac.uk (P.M.); j.reid@qub.ac.uk (J.R.); i.walsh@qub.ac.uk (I.W.); f.wurm@qub.ac.uk (F.W.); helen.noble@qub.ac.uk (H.N.); 2Northern Ireland Kidney Patient Association, Belfast BT9 7AB, UK; william.johnston@kidneycareuk.org; 3Kidney Care UK, Alton GU34 1EF, UK; 4Center for Arts in Medicine, College of the Arts, University of Florida, Gainesville, FL 32611, USA; jlee@arts.ufl.edu; 5South Eastern Health and Social Care Trust, Belfast BT16 1RH, UK; alistair.mackenzie@setrust.hscni.net

**Keywords:** end-stage kidney disease, arts, COVID-19 pandemic, mental health and wellbeing

## Abstract

Patients living with end-stage kidney disease (ESKD) have been seriously impacted by the COVID-19 pandemic. As these patients are considered extremely clinically vulnerable, they were advised to ‘shield’ at home, with limited face-to-face contact and support for the duration of the pandemic. Living with ESKD impacts heavily on patients’ mental health and wellbeing, and this extended period of isolation and loneliness is likely to have a further negative effect on patients’ mental wellbeing. The Renal Arts Group (RAG), Queen’s University Belfast, aims to improve the quality of life of those living with ESKD and the extended renal community through engagement with the arts. We developed an initiative, funded by the Economic and Social Research Council, and carried out an evaluation. The initiative included a programme of online arts-based activities that built upon the work of RAG and provided mental wellbeing support for patients who faced an extended, lonely period of self-isolation. We worked with experienced arts practitioners to identify appropriate activities and developed five workshops and tutorials that were delivered online. We received positive feedback from participants who found the activities to be enjoyable, beneficial to their mental wellbeing and were interested in undertaking further activities online. We conducted interviews with the arts facilitators and identified three themes for consideration when developing online arts activities for the renal community. Participants reported that the activities benefited their mental wellbeing, were enjoyable and provided an opportunity to meet others with shared interests. The arts facilitators reported experiences around accessibility, audience engagement, impact on health and wellbeing and facilitator experience, that should be considered when developing online arts activities for the renal community. This evaluation will inform future work in this area, and the arts tutorial videos developed as part of this project will remain available online for members of the renal community to access.

## 1. Introduction

Patients living with kidney disease that have reached end-stage kidney disease (ESKD) have been seriously impacted by the COVID-19 pandemic [1]. Kidney disease is categorised into five stages. Each stage is defined by the estimated glomerular filtration rate (eGFR) and the percent of kidney function. The fifth stage, or end stage, is the final stage of chronic kidney disease when the kidneys can no longer adequately filter a person’s blood [2]. ESKD is a lifelong condition as there is no cure. Patients with ESKD require renal replacement therapy (RRT) [3] to substitute the normal blood-filtering function of the kidneys. RRT comprises several modalities: haemodialysis (HD), peritoneal dialysis (PD) and renal transplantation [4]. Additionally, included is conservative care, a form of palliative care where the patient decides not to undergo any form of RRT. For most patients with ESKD, in-hospital haemodialysis is the most appropriate option, and patients must attend three times a week for approximately four hours of dialysis. Patients are restricted to a bed, and have their blood removed and filtered through a dialyser. This process must continue every week for the rest of their lives, or until they receive a kidney transplant; however, many patients are unable to be transplanted due to additional health issues. Those who are transplanted require medications to help prevent rejection of the transplanted kidney. These immunosuppressants work by suppressing the immune system leaving patients at risk of infections and further ill health [5]. Living with kidney disease impacts heavily on the mental health and wellbeing of patients [6] and many experience symptoms of depression and anxiety [7].

Due to the nature of their disease, patients with ESKD, including those who are transplanted, are considered extremely clinically vulnerable and where possible were advised to ‘shield’ at home for much of the duration of the pandemic to reduce exposure to COVID-19. This included limited face-to-face contact with family, friends and support networks for an extended period of time [8]. This extended period of loneliness and isolation, with limited face-to-face contact with family, friends and support networks, was likely to have a further negative effect on mental health and wellbeing [9].

Utilising the arts to promote health and wellbeing has gained increased recognition and support since the publication of *Creative Health: The Arts for Health and Wellbeing* report [10], that outlines the benefits of the arts on psychological, social and physical health and wellbeing. The World Health Organisation report on the evidence for arts and health [11] highlighted the wide variety of benefits on health and wellbeing for people living with chronic illnesses such as kidney disease, and recent work in this area has identified the potential for arts interventions to improve the mental wellbeing of patients undergoing haemodialysis [12]. In previous studies, patients engaging in arts activities while undergoing haemodialysis treatment reported an increased sense of accomplishment and control, and reduced anxiety describing the activity as ‘relaxing’ and ‘enjoyable’ [13]. Patients also reported beneficial outcomes including increased self-esteem and sense of purpose [12]. Due to the global COVID-19 pandemic, there has been a lack of opportunities and access for renal patients to engage with the arts due to the requirement for the clinically vulnerable to ‘shield’ at home, along with the closure of museums, galleries and arts centres, furlough of arts staff and restricted access to renal units for non-clinical staff such as art therapists or arts facilitators.

The Renal Arts Group (RAG) at Queen’s University Belfast was formed in 2016 as a collaboration between patients with kidney disease, carers, clinicians, academics and artists to develop a programme of research with the ultimate aim of improving the physical and psychological quality of life of those living with kidney disease through the medium of the arts [14]. The group aims to give a voice to patients with kidney disease, increase public engagement and improve social impact through interdisciplinary collaboration between academics, researchers, healthcare staff, policy makers and service users. Through our monthly meetings, the Renal Arts Group recognized that our group members were struggling with the continued isolation, lack of mental wellbeing support and reduced opportunities for arts engagement due to pandemic restrictions.

The group sought to create opportunities for patients and the wider renal community to participate in arts activities during the pandemic, and to encourage engagement with a creative practice to support patients’ mental health and wellbeing. We also wanted to create opportunities for patients to connect with others with a shared interest in the arts. As it was not possible to access arts activities in-person during this time, we started to think creatively about alternative ways to deliver arts activities.

We planned a project presented below using the *Evaluating Community Projects: A Practical Guide* [15] following the structure of reviewing the situation, gathering evidence, analysing evidence and making use of what we have found, while adhering to the key principles of evaluation.

### Reviewing the Situation

The Renal Arts Group responded to a funding call from the Economic and Social Research Council to develop initiatives that responded to the wide-ranging impacts caused by the COVID-19 pandemic. After undertaking a consultation with our group members, we proposed an initiative that aimed to develop a series of virtual arts-based activities that built upon the work of RAG and supported the mental health and wellbeing of patients facing an extended, lonely period of self-isolation. As the initiative aimed to create engaging online arts activities to support renal patients, we approached arts facilitators who either had personal experience of working with and supporting renal patients, or of delivering arts activities within a healthcare setting. This meant that our facilitators were sensitive to the needs of their audiences’ unique health and wellbeing requirements and pitched their activities accordingly. We planned to develop a series of online arts workshops and tutorials, which could also be shared with the wider renal community to offer an opportunity to engage in a creative practice.

## 2. Method

### 2.1. Participants

We identified two distinct groups of people to take part in the project; those who facilitated the arts activities (referred to as ‘arts facilitators’) and those who voluntarily engaged with the arts activities (referred to as ‘participants’). This group included renal patients, carers, family members and healthcare staff. The arts facilitators (*n* = 5) were responsible for developing and delivering an art activity, following consultation with members of the project team. Participants numbers were collected from those who voluntarily responded to the feedback survey after taking part in an activity (*n* = 24), which is a small percentage of the total number of participants who watched the tutorial videos or took part in the Zoom workshops (*n* = 1214). Of the participants who responded to the survey, 61% of this group identified themselves as living with ESKD or another chronic health condition and a further 4% identified themselves as carers or family members of a person living with a chronic health condition. We are confident that the project’s target group (people living with ESKD, chronic conditions and their carers/family) comprises a sufficient percentage of the participants’ total responses as to be representative.

### 2.2. Gathering Evidence

After undertaking consultation with each arts facilitator, we coordinated five arts activities to be developed for online delivery: creative writing, printmaking, drawing, songwriting and guitar. The facilitators were given the option to deliver their activity either as a live Zoom workshop which participants could join on a specific date and time, or a pre-recorded tutorial video which guided participants through a creative process. All the activities were developed as introductory sessions; therefore, the participants could confidently join the session or undertake the activity without having prior experience or knowledge of the art form. We planned to hold one activity per month between October 2020 and March 2021. The tutorial videos were made available on the Renal Arts Group website [16] and Facebook page [17], Kidney Care UK’s website [18] and were also featured on the Kidney Beam online platform [19], which provides movement, education and wellbeing support for kidney patients. The workshop sessions were held on Zoom, with participants signing up for a session and then accessing the workshop via an emailed link. All the activities were promoted on social media, by our charity partners and through the Northern Ireland Healthcare Trust renal network.

The online arts activities included:Introduction to Printmaking with local Artist and outreach facilitator Ruth Osborne [20]. The facilitator produced 3 videos introducing the concept of printmaking, the materials required and the techniques used to create an artwork.Introduction to Drawing and live drawing workshop with Dr Claire Carswell, Artist, Research Associate and Mental Health Nurse [21]. The facilitator produced a video introducing the materials and techniques required for drawing and led a sketching workshop via Zoom.Introduction to Blues Guitar with Ian Walsh, Consultant Surgeon and Senior Lecturer at Queen’s University Belfast [22]. The facilitator produced a video introducing the basics of how to play a piece of Blues music on a guitar.Introduction to Creative Writing with award-winning playwright and producer Shannon Yee [23]. The facilitator led a Zoom workshop introducing participants to the process of creative writing, including exercises and techniques to practice their skills.Introduction to songwriting with Daniel O’Rourke, transplant recipient and singer/songwriter [24]. The facilitator produced a video introducing his approach to songwriting, how to structure a song and put words to music.

### 2.3. Data Collection

Demographic information was collected relating to the participants’ health status (Table 1), and we collected additional demographic information for the arts facilitators regarding age, gender and previous experience of facilitating groups (Table 2).

### 2.4. Measures

In order to collect data from participants who had engaged with the arts activity, we identified an online survey as the most appropriate instrument to gather participant data and qualitative feedback. The survey was devised with the wider research team which included patients, carers, healthcare professionals and representatives from renal charities (Appendix A). The aim of the arts activities was to be enjoyable, engaging and accessible for renal patients and the wider renal community, and that participants would feel that taking part in the activities was beneficial for their mental health and wellbeing. We also wanted to gain feedback from the participants about how they felt taking part in online arts activities compared with taking part in-person, and if they would be interested in taking part in more online arts activities in the future. The questions in the feedback survey reflected these aims, and invited participants to provide qualitative feedback about how enjoyable they found the activity and if they felt taking part had any impact on their mental wellbeing. Participants were invited to complete the survey via an onscreen message at the end of each video tutorial, or would be invited by the project co-ordinator via email if they had taken part in a Zoom workshop. Completing the survey was entirely optional for participants.

Additionally, we were interested in the arts facilitator’s experience of developing and delivering the online arts activities, and the research team concluded that an interview to reflect on the experience would be most appropriate instrument to gather data from the arts facilitators. The interview schedule (Appendix B) was also supported by the research team, in line with a review of the literature [25,26]), to identify appropriate questions that would explore the facilitators’ experience of delivering an arts activity online. All interviews were conducted online, digitally recorded and transcribed verbatim for analysis. 

### 2.5. Data Analysis

As we wanted to explore the experiences of the facilitators an inductive thematic approach to analysis was adopted. Analysis of the facilitator interviews was carried out using a thematic analysis framework developed by Miles and Huberman [27]. This entails the systematic search of themes and repetitions emerging from the collected data, through the processes of data reduction to select and code data to identify recurring patterns, data display to organise the coded data into a matrix of categories, and drawing and verifying conclusions to identify common themes from these categories [28]. We did not use any coding software or programs. The first author (AW) developed the initial codes, and two co-authors (FW and HN) reviewed the coding scheme and categories were generated which led to the generation of themes.

## 3. Results

### 3.1. Analyse the Evidence

We measured the impact of the project by recording the number of participants taking part in the Zoom workshop sessions, and the number of views that each tutorial video received using YouTube, Facebook and Kidney Beam engagement metric reports. At the end of each online activity participants were invited, via a link to an online survey of 10 questions (Appendix A), to provide feedback on the activity, exploring if they enjoyed the activity and felt it was beneficial to their mental health and wellbeing. We collected demographics identifying if responding participants were patients living with ESKD or another renal condition (*n* = 14), living with another chronic condition (*n* = 2), a carer or family member of someone living with a chronic condition (*n* = 1), an academic/researcher/educator (*n* = 6) or a member of the general public (*n* = 3) (Table 1). Participants could identify as multiple categories if applicable.

To date the arts activity tutorial videos have received over 1200 views, and two live workshops were held via Zoom for drawing and creative writing with a total of 27 participants attending. Very positive feedback has been received from the qualitative survey, with 100% of responding participants rating the arts activities either 4 or 5 out of 5 (overall score 4.71 out of 5) when asked how much they enjoyed the arts activity.


*“Something new for me to try as I enjoy arts and crafts of various kinds. Everything else around is zoned out, I feel in a peaceful place/space as all my mind/concentration is in my art*
*.”*



*“With my hectic family life and uncertainty of the global pandemic this was wonderful to see and be reminded that art is therapeutic.”*



*“It was a positive video and encouraging for anyone that has always thought about doing art, but never had the confidence to start. This will give those people a boost of encouragement.”*


A total of 91% of participants rated the activity as 4 or 5 out of 5 (overall score 4.5 out of 5) as being beneficial for their mental wellbeing.


*“I really enjoyed the session and wish it could have lasted all day. I didn’t realise, but the session felt almost therapeutic, getting thoughts out of my head onto paper and being creative. For the first time in a long time, I felt like I was thinking in lockdown, not clock watching the hours and days away or having a binge on Netflix.”*



*“It felt really good for the first time being part of a group online, all interested in the same thing, sharing that passion.”*


A total of 100% of participants expressed an interest in taking part in online arts activities in the future.


*“I enjoy online activities as I can take them at my own pace which I find beneficial (depending were I am at in my own headspace and in myself).”*



*“Face to face would be more interactive but this is a wonderful initiative in a difficult time.”*


As part of the evaluation, the five arts facilitators were invited to take part in a structured reflective interview (Appendix B) about the activity that they delivered and three common themes were identified: Accessibility, Audience Engagement and The Impact on Health and Wellbeing and Facilitator Experience–Challenges and Opportunities.

### 3.2. Accessibility

The first theme, ‘Accessibility’, related to the facilitator’s consideration of how to best communicate the activity through the choice of online delivery method. Three chose a pre-recorded tutorial video, one chose a live workshop delivered via Zoom, and one facilitator chose to do both. The facilitators demonstrated a high level of thought and consideration in planning how to best communicate their arts activity using an online delivery method, taking into account their diverse audience who would taking part within their own home rather than a more formal environment. They recognised that in delivering their arts activity online they needed to give flexibility to take part in the activity in a way that suited the individual.


*“…so I had to figure out an exercise that was going to work, but work for multiple different people on their own, in their house.”*
(F1) (F = Facilitator—refer to Table 2)


*“Print making requires some specific materials, they could watch the video, source materials, and then re-watch it and have a go in their own time.”*
(F2)

The facilitators recognised that many participants may not previously taken part in a similar activity and might need additional support and reassurance. As such they planned their activity as an introductory session that aimed to build confidence and provide an opportunity for the audience to take part in something new. They also recognised that although online delivery may be a new approach for many, there was potential for a wider reach and greater outreach than delivering in-person sessions.


*“… the goal was to help them just to get a sense of the basics, to empower them … lots of times there’s what I feel is an incorrect prejudice, or idea that being a writer is out of reach for anybody.”*
(F3)


*“You do lose a lot by online delivery, but […] I think you do gain a lot too, because there is that flexibility and there is a bit more outreach. If you are doing stuff in person, you are quite nuanced in who you have in the room with you. whereas there is the capability to really spread the word a lot wider”*
(F4)

The facilitators reflected on their activity and considered how they had adapted their approach to suit the online nature of delivery, and what changes they might make if they were to undertake online arts activities in the future.


*“It was much more about distilling the activity down to the core components. And making it as accessible as possible.”*
(F2)


*“I would like to have more input from the people who are participating in it, ahead of time, so I can tailor what we are doing to their interests and what they want to get out of it.”*
(F1)

### 3.3. Audience Engagement and the Impact on Health and Wellbeing

The second theme, ‘Audience Engagement and The Impact on Health and Wellbeing’, related to how participants would take part in the activity and how participation might improve their health and wellbeing. The facilitators had considered how the audience would approach and participate in the activity, recognising that participants might be anxious or self-critical about their ability and aimed to make the activities encouraging and accessible.


*“The video tutorial was essentially breaking down all of these preconceptions that we have about art and drawing and saying, look, there’s no rules. You can do it whatever way it works for you. You can draw that way. You don’t need to set any limits on that*
*.”*
(F1)


*“…because I pitched it … accessible to everybody. Everybody can write. And presenting it as, these are tools that you can have in your writer’s toolkit that you can then apply throughout your writing journey.”*
(F3)

In addition, it was considered how the activities could provide an opportunity for escape or respite from participants’ ongoing health issues, and could alleviate stress through the process of creating art. The additional isolation felt by renal patients during the extended period of shielding due to the pandemic restrictions was also recognised, and activities provided a sense of community for patients taking part, as well as a safe space to share their experience with others in similar situations.


*“…so you can have a different component to your life that isn’t about illness. It’s not about caring for someone. It is this thing for yourself that you can be proud of and that you can attribute meaning to that is separate from disease.”*
(F1)


*“I considered my participants in my planning is that some of the writing might have touched on or brought up some unpleasant memories of trauma or of medical experiences that were unpleasant…it was also a safe space in which people felt a sense of community, even though they didn’t really know each other. But that communal… that camaraderie from struggle and a shared experience of chronic illness or transplant or dialysis was a strength of the group.”*
(F3)

### 3.4. Facilitator Experience–Challenges and Opportunities

The third theme, ‘Facilitator Experience–Challenges and Opportunities’, related to the facilitators personal experience of delivering their arts practice in a new way. Challenges faced by the facilitators were identified and included issues with technical equipment or managing the impact of online delivery on participant engagement.

*“I encountered a few technological difficulties! But that’s more to do with me than the actual process, just making sure that light levels and fixed focus when filming and things like that”*.(F2)

However, the facilitators did express that they had learned new skills from the experience of delivering an arts activity online, in particular video editing, using online technologies and the ability to adapt their usual approach to suit the delivery method. Several facilitators had been encouraged to create more online arts activities as a result of this project.


*“A year ago I would have thought, you can’t do this. Particularly arts and especially music, you just can’t do this. I pivoted completely in my views about that”*
(F4)

The facilitators also reflected on their own experiences of the pandemic, and how it has affected their creative practice. Their responses revealed negative impacts of the pandemic and associated restrictions, including the inability to deliver arts workshops, take part in live performance, as well as increased stress due to increased risk to health, additional childcare responsibilities and increased workload, which have led to lack of time and creativity.


*“You don’t have the mental space to be as creative as you’d like to be. It’s just inspiration is really difficult to come by because every day is a little bit like Groundhog Day and you are doing the same stuff. And you are just getting through to the end.”*
(F5)

The facilitators did recognise positive aspects that have stemmed from the pandemic restrictions, including spending more time at home with family, increased opportunities for online engagement without geographical or travel restrictions and how arts engagement has supported their own wellbeing.


*“I work and then I draw. And that’s my life. And I am quite happy with that. It helped me get through the pandemic.”*
(F1)


*“It has bizarrely positively opened up opportunities for me as an artist because people have moved online. So opportunities for workshops, for professional development, and also work have opened up in ways that would not have been available previously.”*
(F3)

## 4. Discussion

The aim of this study was to evaluate a series of virtual arts activities for renal patients and the wider community during the COVID-19 pandemic, which built upon the previous work of the Renal Arts Group [12,29,30,31], provided an opportunity to engage in an enjoyable creative practice and supported the mental wellbeing of those ‘shielding’ due to the pandemic restrictions. We found that participants reported the activities to be enjoyable, beneficial for their mental wellbeing and the group activities provided an opportunity to meet others with shared interests. Through a series of interviews with the arts facilitators we determined three common themes: accessibility, audience engagement and the impact on health and wellbeing and facilitator experience–challenges and opportunities, that should be considered when developing online arts activities for the renal community.

Engagement with the arts has long been reported as having beneficial outcomes for patients living with chronic illness. Reynolds and Lim [32] identified the contribution of artmaking to the promotion of subjective wellbeing in patients living with cancer, whereas a study of rural women living with chronic conditions, including diabetes, cancer, arthritis and multiple sclerosis, recognised that engaging in a creative activity significantly improved their ability to cope with chronic illness and increased their overall sense of wellbeing [33]. A study of patients living with chronic obstructive pulmonary disease found that engaging in a community singing group, in addition to positive effects for physical health, participants reported improved mental wellbeing and social support from other patients with shared health experiences [34]. The studies support our findings that engaging in a creative practice or activity has beneficial outcomes for patients living with chronic health conditions. As was found in a previous arts-based intervention for renal patients [12] participants responded positively to the activities and expressed a strong interest in taking part in future arts activities, and the evaluation of this project contributes to the growing body of evidence on the health benefits of taking part in leisure activities such as the arts [35].

### 4.1. Make Use of What You Have Found Out

Although the online arts activities have received very positive feedback, there are a number of factors to consider which may have impacted the progress of the initiative and should be reflected upon going forward. Due to the changing nature of the pandemic restrictions during the timeline of the project, there was increased access to in-person arts activities in the later months of 2021 with most museums, galleries and arts centres reopening to the public, offering workshops and classes led by an artist or facilitator; however, many had reduced programmes and in-person activities. Delivery of arts activities within renal units remained restricted with only healthcare staff able to access clinical zones within hospitals. The advice for ESKD and transplant patients was varied across the UK and many patients were not comfortable increasing their contact with others and continued to ‘shield’. Although some members of the renal community may have experienced a reduced need for online arts activities, the initiative continued to provide an opportunity for creative engagement for those who did not wish to increase their contact with others.

A factor which may have impacted the delivery of the initiative is the arts facilitators’ lack of experience in delivering online sessions. Although the facilitators faced a range of challenges when tackling technical issues, and considering how participants would engage with the activity, most reported that taking part in the initiative allowed them to gain knowledge and experience in online delivery, and would be encouraged to deliver further online arts sessions in the future.

The project also experienced some difficulties in encouraging participants to complete the post-activity survey. There was a greater rate of response from participants taking part in the Zoom workshop sessions than those who watched the tutorial videos. This may be because workshop participants were required to ‘sign up’ to each session via email request, therefore a follow-up email could be sent directly to the participant as a reminder to complete the survey. Participants watching the online tutorials did not have to submit any contact details in order to watch the videos, and therefore could not be emailed with the same reminder. Another issue to consider when reflecting on the initiative is the need to establish a comprehensive baseline in order to measure any change in participants’ responses. Ideally the project should have included a pre-activity survey for participants that would allow for changes in responses to be measured; however, due to the informal nature of activities we did not want to ask participants to undertake too much additional work. In terms of limitations, the scope of the project was small due to resource constraints and pace of development; however, we hope that the findings can be viewed within the wider context of the field with the potential for further exploration in future.

We also experienced an unforeseen issue in that participants expressed interest in taking part in activities beyond what was able to be offered within the scope of the initiative, with requests for additional workshops and activities on a more regular basis. The level of interest from participants reflects positively on the activities that we were able to offer, and highlights the need for further investment in providing arts interventions for the renal community.

### 4.2. Share Your Findings with Others

In evaluating the project, we have identified that members of the renal community are interested in taking part in online arts activities, and that participants recognized the potential for the arts to benefit their mental health and wellbeing. The initiative unlocked the potential of online delivery to reach the renal community and increased access to arts activities during the period of restrictions imposed by the pandemic. The delivery method provided an opportunity for patients and the extended renal community to engage with the arts in a safe and familiar space, allowing participants to connect to others living with similar conditions and experiences during the workshop sessions.

The findings from our initiative will be shared with our project partners, including Kidney Care UK, Northern Ireland Kidney Patients Association and Kidney Beam. It is hoped that the findings from this initiative will inform future work in this area, and the arts tutorial videos developed as part of this project will remain available on the Renal Arts Group and Kidney Care UK websites and Kidney Beam online platform for members of the renal community to participate and engage with in the future. There is also potential for the videos to be included in arts interventions programmes offered to patients during dialysis sessions.

The impact of the initiative is already being seen in increased engagement with the arts within the renal community. The Young Adult Kidney Group [36], a support group for young people living with renal conditions and supported by Kidney Care UK, has invited one of the arts facilitators from the initiative to provide an online creative writing sessions for participants as part of their virtual activity weekend. The workshop will be specifically tailored to the groups’ needs, providing an ongoing opportunity for engagement and participation with the arts.

## 5. Conclusions

During the ongoing course of the pandemic there has been a huge increase in the provision of online arts activities, with artists, arts organizations and museums adapting in-person activities for online delivery, resulting in much wider access to creative activities available online, with many more individuals accessing online arts resources [26]. This shift in delivery method could provide a greater opportunity for the renal community to access and explore other creative practices online, opening up the potential for increased arts engagement for renal patients, carers, family members and healthcare staff. Working with experienced arts facilitators ensured that the arts activities developed as part of this project were planned with participants’ unique health needs in mind, and were delivered in an accessible and engaging manner that was appropriate to be undertaken within participants’ home. We also identified that participants have a strong interest in taking part in more online arts activities across a greater range of creative practices.

## Figures and Tables

**Table 1 healthcare-10-00260-t001:** Participant Demographics.

Participant Health Background	Percentage of Responding Participants
Living with ESKD or other renal condition	53%
Living with another chronic condition	8%
A carer or family member of someone living with a chronic condition	4%
Academic/researcher/educator	23%
General Public	12%

**Table 2 healthcare-10-00260-t002:** Arts Facilitator and Activity Information.

Facilitator	Age	Gender	Activity Delivered	Delivery Method	Previous Experience Facilitating Groups
F1	20–29	Female	Drawing	Video tutorial + Zoom workshop	Yes
F2	30–39	Female	Printmaking	Video tutorial	Yes
F3	40–49	Female	Creative Writing	Zoom workshop	Yes
F4	50–59	Male	Guitar	Video tutorial	Yes
F5	20–29	Male	Songwriting	Video tutorial	Yes

## Data Availability

The data presented in this study is available on request from the corresponding author. The data is not publicly available as this was not a requirement of the funding.

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
