# Peer review of "Evaluation of a Programme of Online Arts Activities for Patients with Kidney Disease during the COVID-19 Pandemic"

_healthcare, 2022, doi:10.3390/healthcare10020260_

Round 1

Reviewer 1 Report

The authors are to be congratulated on their careful attention to all reviewer requests. This article now appropriately reports the background, findings, and process of a very valuable study that will contribute useful information for those experiencing and assisting people with End Stage Kidney Disease. The only suggestion I would still make is to place the appendices after the References list...but that could be relegated to editing level. Well done on producing a great paper!

Reviewer 2 Report

This paper deals with a timely topic of relative relevance in this Covid pandemic time. The findings, however, are not very substantial as they only confirm some intuitions, which seem to be rather obvious. The paper as a whole reads like a plea for the implementation of a programme for online arts activities, but the text does not meet strong academic standards. The methodological section is rather weak, and the theoretical background is almost totally lacking. The paper should have been much stronger, if the findings would have been positioned within a theoretical framework that explains why there is need of such a programme, and why such a program could be beneficial to some extent. I do not think that the paper, as it stands now, is sufficiently strong to be accepted in an academic journal. In order to provide some constructive-critical remarks, I list below some general remarks and detailed comments.

General remarks

  • The subject matter is timely and has some relevance, but the content as whole is not very substantial.
  • The style of writing is very circumstantial, and seems more like a chronicle than a scholarly article.
  • Some passages are very anecdotical rather than being systematic descriptions of an experimental design.
  • The course of the online sessions is not described in detail. It is not clear for the reader how they have been implemented. There are, of course, the links to the online arts activities on the internet, but the paper should be self-sufficient in the sense that a short description of these sessions should be inserted also in the main paper.
  • The methodological description is very weak. What is the major research question, the aim of the article, the methods used, the major contribution, and conclusions and perspectives? Not all of these issues have been fully addressed.
  • Some seminal papers for grounding the methodology are rather old. Some updated references could be helpful here.
  • The academic standards of the paper or rather low. Why so many authors for such a simple paper? What is the contribution of each of them?
  • It seems that the whole project is still in a tentative and exploratory state, which gives the impression that it could be considered as a pilot study for a more in-depth elaboration at a later stage.
  • The project is very strongly connected to the organizing health centre. which gives it a very personal touch. More generalization is needed to meet academic standards. Now, it reads more as a kind of advertisement rather than a scholarly description.
  • The reference list is very limited with very few articles that can serve as a broader theoretical background.

Detailed comments

  • line 132: The number of arts facilitators is very limited (n = 5). What about inter-rater reliability for the assessment of the results?
  • line 172: add a bullet to the bullet list
  • line 205: a review of the literature. Which literature? No references are given.
  • line 213: reference to Miles and Huberman is a rather old reference (1994) to be used as the starting reference for the used methodology. Some updated references could be helpful here to show that this method is still used today.
  • line 219: what about the inter-rater reliability (Cronbach alfa)?
  • line 235: provide the number of subjects (n = ?), not merely the percentages
  • line 398 ff: this is interesting material, but some supporting literature is lacking
  • line 421: what is meant with “in-person” arts activities?
  • line 445: why was the pre-activity survey not done? This is a methodological weakness;
  • line 465 ff: this information is not relevant for an academic paper

Round 2

Reviewer 2 Report

Thank you for having addressed my main concerns and questions. I think the paper can now be accepted for publication.

This manuscript is a resubmission of an earlier submission. The following is a list of the peer review reports and author responses from that submission.

Round 1

Reviewer 1 Report

The corrections that have been made on the basis of the reviewers' suggestions have substantially improved the article, which is now better than before.

Reviewer 2 Report

A Method (I recommend ‘Method’ rather than ‘Methods’) section has been created but clarification of specific detail still needs to be defined.

Please create a designated ‘Participants’ section. Please explain specifically who the participants are and give the number of each participant group in this section. There appear to be two distinct larger groups of participants: people engaging in the activities (shown in Table 1), and people facilitating the activities (shown in Table 2). In the text pertaining to the participants please clearly define each participant group. I believe it would be helpful too, in view of the multiple sub-groups comprising the cohort responding to the Appendix A survey, to express confidence that the target group (people with serious health issues-61%) comprises a sufficient percentage of that cohort’s total responses as to be representative.

Please create a discrete section titled ‘Measures’ and briefly detail each of the measures, their purpose and the participant group to which they pertained, and upon which aims or previous research they were developed.

Clarity around who the participants actually are, and for whom the measures are designated then, are still needed…for example, in line 183 this text appears: ‘We measured the impact of the project by recording the number of participants taking part in the Zoom workshop sessions’…to which participant group does this pertain?... and around line 185,  ‘At the end of each online activity participants were invited…’. Specifically which participant group was invited? Please be specific. In short, supply the appropriate details in the clear sections and be specific about who these groups are.

In terms of analysis, Miles and Huberman’s (1994) method is perfectly acceptable, but briefly explain what the steps are please. Bear in mind that all these details would enable other researchers to repeat the process if they wished to conduct a similar study. Overall, the detail provided in the Method section (and sub-sections) should enable this.

Finally, the aim as stated in line 328 requires clarity also: ‘The aim of this evaluation was to develop a series of virtual arts activities for renal patients and the wider community during the Covid-19 pandemic…’

I suggest that this would be better aligned with the title of the paper as follows: The aim of this study was to evaluate a series of virtual arts activities for renal patients and the wider community during the Covid-19 pandemic…’

Overall, while clearly a valuable study with useful findings, this paper needs to be made clear and specific in detail.

Reviewer 3 Report

Dear Authors,

The information added to the manuscript is not considered to resolve the underlying design problems of the research. The study has methodological flaws that make it difficult to obtain solid evidence.